# From Interconnection between Genes and Microenvironment to Novel Immunotherapeutic Approaches in Upper Gastro-Intestinal Cancers—A Multidisciplinary Perspective

**DOI:** 10.3390/cancers12082105

**Published:** 2020-07-29

**Authors:** Giulia Accordino, Sara Lettieri, Chandra Bortolotto, Silvia Benvenuti, Anna Gallotti, Elisabetta Gattoni, Francesco Agustoni, Emma Pozzi, Pietro Rinaldi, Cristiano Primiceri, Patrizia Morbini, Andrea Lancia, Giulia Maria Stella

**Affiliations:** 1Department of Medical Sciences and Infective Diseases, Unit of Respiratory Diseases, Istituto di Ricovero e Cura a Carattere Scientifico (IRCCS) Policlinico San Matteo Foundation and University of Pavia Medical School, 27000 Pavia, Italy; accordino@gmail.com (G.A.); sara.lettieri01@universitadipavia.it (S.L.); 2Department of Intensive Medicine, Unit of Radiology, IRCCS Policlinico San Matteo Foundation and University of Pavia Medical School, 27000 Pavia, Italy; c.bortolotto@smatteo.pv.it (C.B.); a.gallotti@smatteo.pv.it (A.G.); 3Candiolo Cancer Institute, Fondazione del Piemonte per l’Oncologia (FPO)-IRCCS-Str. Prov.le 142, km. 3,95, 10060 Candiolo (TO), Italy; silvia.benvenuti@ircc.it; 4Department of Oncology, Azienda Sanitaria Locale (ASL) AL, 27000 Casale Monferrato (AL), Italy; bettygattoni@gmail.com; 5Department of Medical Sciences and Infective Diseases, Unit of Oncology, IRCCS Policlinico San Matteo Foundation and University of Pavia Medical School, 27000 Pavia, Italy; f.agustoni@smatteo.pv.it (F.A.); e.pozzi@smatteo.pv.it (E.P.); 6Department of Intensive Medicine, Unit of Thoracic Surgery, IRCCS Policlinico San Matteo Foundation and University of Pavia Medical School, 27000 Pavia, Italy; p.rinaldi@smatteo.pv.it (P.R.); c.primiceri@smatteo.pv.it (C.P.); 7Department of Diagnostic Medicine, Unit of Pathology, IRCCS Policlinico San Matteo Foundation and University of Pavia Medical School, 27000 Pavia, Italy; p.morbini@smatteo.pv.it; 8Department of Medical Sciences and Infective Diseases, Unit of Radiation Therapy, IRCCS Policlinico San Matteo Foundation and University of Pavia Medical School, 27000 Pavia, Italy; a.lancia@smatteo.pv.it

**Keywords:** immunotherapy, genetics, gastric cancer, esophageal cancer, multidisciplinary

## Abstract

Despite the progress during the last decade, patients with advanced gastric and esophageal cancers still have poor prognosis. Finding optimal therapeutic strategies represents an unmet need in this field. Several prognostic and predictive factors have been evaluated and may guide clinicians in choosing a tailored treatment. Data from large studies investigating the role of immunotherapy in gastrointestinal cancers are promising but further investigations are necessary to better select those patients who can mostly benefit from these novel therapies. This review will focus on the treatment of metastatic esophageal and gastric cancer. We will review the standard of care and the role of novel therapies such as immunotherapies and CAR-T. Moreover, we will focus on the analysis of potential predictive biomarkers such as Modify as: Microsatellite Instability (MSI) and PD-L1, which may lead to treatment personalization and improved treatment outcomes. A multidisciplinary point of view is mandatory to generate an integrated approach to properly exploit these novel antiproliferative agents.

## 1. Introduction

The definition of upper gastrointestinal (GI) cancers essentially refers to gastric and esophageal tumors. The latter, including both squamous cell carcinoma and adenocarcinoma histologies represents the nineth cause of cancer death worldwide and nearly 40% of patients present with metastatic disease at diagnosis [1,2,3]. The median 5-year survival rate is 47% in case of early stage diseases whereas it decreases to 25% in locally advanced and to 5% for metastatic disease, respectively. Gastric cancer (GC) remains one of the most common and deadly cancers worldwide. Over one million cases of GCs are diagnosed every year around the world. It is the 5th most diagnosed cancer in the world [4]. The epidemiology of stomach cancer harbors substantial geographical heterogeneity and the 5-year survival rate is around 20%, with peaks of about 65% in Japan and 71.5% in South Korea, due high number of diagnosis in early stage disease revealed by massive population screening programs [5]. The geographic variations are mainly related to differences in environmental factors such as dietary patterns and salt intake, the prevalence of *Helicobacter pylorii* (*H.P*) infection and the virulence of strains, as well as host factors [6]. Overall patients affected by resectable cancer can undergo surgery and perioperative therapy with potentially curative purposes. However, most of GC diagnoses are performed in stage III or IV disease and patients are candidates only to palliative chemotherapy. In metastatic diseases, 5-year survival rate is poor with a median overall survival (OS) lower than 12 months [7]. Genomic and proteomic expression profiles of oncogenic signaling pathways unveiled different molecular subtypes of gastric and gastro-esophageal cancers, characterized by specifically targetable markers [8,9,10]. The most relevant example regards the HER2 inhibitor trastuzumab, a monoclonal antibody that binds to the extracellular domain of the receptor, which is now approved in United States and Europe as the first-line treatment in combination with conventional chemotherapy for HER2-overexpressing locally advanced or metastatic GCs (about 20% of cases [11]) leading to increased overall response rates and survivals [12]. Nevertheless, the introduction of targeted molecules does not result in increased outcome rates and most phase III clinical trials evaluating molecularly designed agents in GC have failed [13]. In this complex landscape, growing evidence supports the routine use of immunotherapy with checkpoint inhibitors in the treatment of upper GI cancers, although their effective role is, still, poorly understood. The main reason is due to the lack of knowledge on how the genetic asset cooperates with the surrounding stroma giving rise to the highly malignant phenotype which defines these tumors. Here we summarize the already available data on the use of checkpoint inhibitors and discuss more recent findings regarding the use of modern immunotherapy, including adoptive cell therapy and vaccines, alone or in combination with conventional drugs. A deep understating of the complex interaction between tumor microenvironment and genetic heterogeneity in this group of tumors, fully requires a multidisciplinary approach that will allow effective and significant clinical results. 

## 2. How to Diagnose and Stage Upper GI Cancers

Primary esophageal cancer (EC) constitutes the majority (more than 95%) of all esophageal malignancies. The two main pathologic subtypes of esophageal cancer are squamous cell (ESCC) carcinoma and adenocarcinoma (EAC). The latter can mimic metastases or direct extension from tumors of lung or breast. Adenocarcinomas (AC) represent more than 90% of gastric cancers; considering tumor localization they are subdivided into true gastric AC and GEJ-AC. Growing evidence documents a shift in the anatomical distribution of gastric cancer, which increasingly originates from the proximal stomach near the junction with the esophagus and in parallel an increase of EAC affecting the lower esophagus [14]. Thus, a significant uncertainty might regard the identification of the primary organ site of adenocarcinomatous transformation. Immunohistochemistry (IHC) is helpful in defining pathologic entities in case of undifferentiated cancers from the upper GI tract [15,16]. The most common secondary malignant lesions are associated to localization of lymphoma and sarcoma; metastatic masses arousing in the esophagus are rare [17]. Table 1 summarizes the main morphological and IHC features of primary upper GI cancers. In the case of esophageal adenocarcinoma lesion, differential diagnosis to establish the putative primary origin takes into consideration the lung, in which cells frequently express the thyroid transcription factor 1 (TTF-1), and breast adenocarcinomas, which are generally positive for estrogen receptor (ER), mammaglobin, gross cystic fluid protein and GATA3. On the other hand, ESCCs carry some of the same features of small cells carcinomas which develop in other organs, particularly in the lung and which differentially express common neuroendocrine markers, namely synaptophysin, chromogranin A and CD56/NCAM. Notably, the TTF-1 expression can be found in a proportion of ESCCs as well; thus it cannot specifically indicate the lung only as site of primary tumor growth. 

Comprehensive description of epidemiologic, clinic and pathologic features of upper GI cancers goes beyond the scope of this work and is already available in many published review papers. All the data summarized in Table 2. Once diagnosis of esophageal/gastric cancer is accurately confirmed, disease pathologic classification and staging are required to address patients to the better therapeutic approach. Siewert classification is a widely used anatomic classification of adenocarcinoma of GEJ and it is based on tumor location with respect to the gastric cardia. Three types are described: Siewert type I tumors are adenocarcinomas of distal esophagus, Siewert type II tumors are adenocarcinomas of gastric cardias and Siewert type III tumors correspond to sub-cardial adenocarcinomas of proximal stomach infiltrating the GEJ. The most recent WHO histopathological classification (WHO Classification of tumors: Digestive system tumors 2019) modified the conventional Lauren’s criteria distinguishing gastric cancer into diffuse and intestinal types: diffuse type was reclassified as “poorly cohesive, including signet ring histotype,” while intestinal type was split into architectural types papillary and tubular [21,22]. Previous gastrectomy is a known risk factor for the onset of gastric cancer. The so-called Gastric Stump Cancer (GSC), which occurs in the gastric remnant at least 5 years after the surgery for peptic ulcer, identifies a separate subtype of GC (1.1/7% of diagnosis) which mainly affects men [23,24,25]. The TNM classification represents the most used staging system for upper GI tumors. Details regarding upper GI staging and classification are available as Appendix A [26,27,28,29,30].

## 3. Main Mutational Patterns and Regulatory Networks

### 3.1. Oesophageal Cancer

The genomic landscape of ESCC and EAC have been extensively studied through next generation sequencing (NGS) and computational approaches, even though the understanding of the complex network of its driver genes is far to be fully understood. ESCC and EAC display distinct sets of driver genes, mutational signatures and prognostic biomarkers.

Esophageal squamous cell carcinomas resemble squamous carcinomas of other organs more than they did esophageal adenocarcinomas. The work conducted by Cancer Genome Atlas Research Network revealed that ESCC showed frequent genomic amplifications of *CCND1* and *SOX2* and/or *TP63* genes, whereas *ERBB2*, *VEGFA* and *GATA4* and *GATA6* were more commonly amplified in adenocarcinomas [47]. Inactivation of the tumor suppressor *NOTCH1* gene has been reported in ESCC but not in EAC [48]. Interestingly inactivating mutations clustered in defined geographic areas, being more frequent in those ECSSs which affect North American patients than in those aroused in Chinese population. Moreover, germline mutations in the RHBDF2 gene (17q25) which cause tylosis (focal non-epidermolytic palmoplantar keratoderma) have been reported to be markers of genetic familial susceptibility for the early onset of ESSC [49,50,51].

On the other hand, EAC derives from progressive accumulation of multiple genetic abnormalities and aneuploidy. Comparative analysis show that most mutations found in EAC could be already detected in the matched BE which, - at least under genetic profile - identifies an early phase of malignant transformation [52]. Mutations in the *PIK3CA* oncogene and in the *CTNNB1* gene that encodes for β-catenin are known to occur in BE and changes in several tumor suppressor genes involved in chromatin remodeling, such as *ARID1A* and *SMARCA4* as well as in *TP53* and *SMAD4* are usually found in tissues with high-grade dysplasia and EAC. Oncogene amplification is typically a late event in EAC progression. Coherently, genomes of BE tissues are relatively stable compared to those of invasive tumors, in which almost 40% of the genome is non-diploid. The only common copy number alteration found in BE is 9p loss of heterozygosity (*CDKN2A*) [53,54]. Advanced tumors have an increased copy numbers of several oncogenes (*GATA4*, *KLF5*, *MYB*, *PRKCI*, *CCND1*, *FGF3*, *FGF4*, *FGF19 and VEGFA*) and loss of common fragile sites (*FHIT*, *WWOX*, *PDE4D*, *PTPRD and PARK2*) [55,56]. In conclusion, EACs emerge rather than from the gradual accumulation of tumor-suppressor alterations, from a straighter pathway driven by mutations in *TP-53* gene and subsequent acquisition of oncogene amplifications [57]. In this perspective, EACs strongly resemble the chromosomally unstable variant of gastric adenocarcinoma, suggesting that these cancers could be considered as a single disease entity. However, some molecular features, including DNA hypermethylation, occur disproportionally in esophageal adenocarcinomas. Epigenetic modifications are known to contribute significantly to the pathogenesis of the disease and specific methylation signatures are known to be associated to tumor progression processes and thus emerge as novel actionable markers. Among them, the methylation classifier which encompasses the *TRIM15*, *TACC3*, *SHANK2*, *MCC* and *CDKN2A* gene silencing is differentially reported in the progression from BE to transformed areas and not in normal mucosa [58].

### 3.2. Gastric Cancer

#### Genetic Features 

Gastric cancer is characterized by an extreme molecular heterogeneity, which is defined by the occurrence of multiple genetic and epigenetic alteration in each disease stage. It should be underlined that 3–15% of all diagnosis refer to familial and hereditary gastric cancers, among which hereditary diffuse gastric cancer (HDGC), gastric adenocarcinoma and proximal polyposis of the stomach (GAPPS) and familial intestinal gastric cancer (FIGC). One third of HDGC is attributed to hereditary *CDH1* mutations [59,60,61]. Other hereditary syndromes, such as Lynch syndrome, familial adenomatous polyposis (FAP), Li-Fraumeni, Muir-Torre and Peutz-Jeghers syndromes can occur with gastric involvement as well [62,63,64]. In case of genetic diagnosis, prophylactic gastrectomy might be suggested [65,66]. Within respect to the non-hereditary forms of GC, recent molecular profiling studies have allowed the shift from the conventional histological classification systems to four molecularly-based classification groups: (i) EBV-positive cancers (9–10% of gastric AC) harboring high frequency of *PI3KCA* gene changes (80%), high levels of DNA hypermethylation, mutations in *PTEN*, *SMADA*, *CDKN2A*, *ARIDA* (55%) and *BCOR* (23%) and increased copies of *JAK2*, *ERBB2*, *PD-L1* and *PD-1* genes, (ii) microsatellite unstable (MSI) tumors, accounting for 22% of diagnosis, which mainly arise in women and older patients and frequently carry hypermethylation MLH1 promoter in association with recurrent mutations in the *PIK3CA*, *ERBB3*, *ERBB2* and *EGFR* genes [67,68,69], (iii) genomically stable (GS) tumors (20% of cases, mainly diffuse-type AC) which mostly affect younger subjects and are enriched with recurrent *CDH1* (37%), *RHOA* (15%) and inactivating *ARID1A* gene changes. Fusions involving the RHO-family GTPase-activating proteins CLDN18 and ARHGAP26, have been reported as well [70]; (iv) chromosomal unstable (CIN) subtypes [71] which account for 50% of GCs and harbor extensive aneuploidy, *TP53* mutations (71%) and increased copy number of several genes encoding for receptor tyrosine kinases and their downstream effectors as *EGFR*, *ERBB2*, *ERBB3*, *VEGFA*, *FGFR2*, *MET*, *NRAS/KRAS*, *JAK2*, *CD274*, *PDCD1LG2* and *PIK3CA* [72]. Overall tyrosine kinase receptors (TKR) are among the most frequently altered oncogenes in GC and identify actionable therapeutic targets. A recent genomic study of gastric cancers identified somatic copy number alterations of seven oncogenes involved in tyrosine kinase/MAP-kinase pathways: *KRAS*, *EGFR*, *HER2*, *FGFR1*, *FGFR2*, *MET* and *IGF1R* [73]. 

### 3.3. Targeted-Based Therapeutic Strategies

Although a deep analysis of genetic basis of targeted therapy in GCs falls beyond the scope of this review, some relevant issues are discussed due not only to their clinically relevant role but mainly to their interaction with microenvironment and immune response. A first example regards the blockade of HER2 signaling which has significantly improved the outlook for esophagogastric cancer patients and has allowed the approval of trastuzumab in HER2-positive metastatic gastric/gastroesophageal junction cancers, as first line approach in combination with cisplatin and a fluoropyrimidine (capecitabine or 5-fluorouracil) [12,74]. HER2 is activated most frequently by increased gene copy number, whereas somatic mutations rarely occur [75]. *HER2* gene amplification in GC is associated with higher invasive and proliferative tumor cell capacity [76]. HER2-overexpression is associated with male gender, intestinal type and well/moderate cell differentiation [77]. Anti EGFR antibodies, cetuximab and panitumumab, combined with chemotherapy did not show benefit in overall survival in first line treatment in metastatic gastric patients, as reported in two phase III trials, EXPANDED and REAL 3 [78,79]. The angiogenesis is another target in the therapeutic strategy against some solid tumors like breast, colon and lung cancer and in some instances; it resulted as good target of therapy. In upper GI cancer Bevacizumab with chemotherapy obtained in one phase III trial better overall response rate but failed to gain benefit in overall survival (OS), the primary endpoint of that study [80]. Ramucirumab also, another antiangiogenic drug, combined with chemotherapy compared with chemotherapy alone in phase III trial in metastatic GC patient chemotherapy naive, showed better progression free survival (PFS) (5.7 vs. 5.4 months) in absence of significant OS improvement [81]. In metastatic patients progressed after platinum-based chemotherapy, ramucirumab plus paclitaxel gained benefit in OS, as reported in RAINBOW, phase III trial [82]. Also used as single agent, in a phase III double blind study, in metastatic patient progressed after standard first line chemotherapy, ramucirumab obtained benefit in OS and good tolerability (REGARD STUDY) [83]. Inappropriate activation of MET signaling occurs in a several cancer types, including gastric cancer and promotes tumor cell growth, survival, migration and invasion, namely the Invasive growth genetic program which is involved tumor spreading and metastatic growth [84,85]. Amplification/overexpression of the HGF-receptor MET rather than mutated gene can activate receptor tyrosine kinase [86,87]. MET overexpression/amplification is more common in intestinal-type GC and reported in diffuse GC [88]. Notably, a cross talk between amplified MET and EGFR, HER2 and HER3 has been described and can establish a signaling network, allowing constitutive PI3K/AKT cascade activation [89]. This observation suggests robust rationale for combinatorial therapeutic approaches against MET and EGF receptor family, at least in metastatic GCs [90,91]. DNA repair *BRCA1/2* genes mutations are implicated in defective DNA repair processes and are known to be associated to the susceptibility towards hereditary breast and ovarian cancers and can occur in other sporadic cancers, among which gastric cancers. *BRCA1/2* mutations are found in 15% of GCs and are associated with poor patient survival [92]. Overall, *BRCAness*—the phenotypic condition that characterizes some cancers with carry defective caretaker gene functions—is associated to high sensitivity to the antitumor agents which cause double strand breaks of DNA, such as platinum [93]. However this condition suggests that GCs might benefit from either platinum therapy or poly (ADP-ribose) polymerase (PARP) inhibitors, a family of nuclear proteins with enzymatic, scaffolding properties and recruiting ability for DNA repair proteins and have been already tested in gastric cancer. However, the first results with PARP inhibitors did not provide encouraging results in metastatic gastric cancer according to a phase 3 study (GOLD), in which olaparib was used in combination with paclitaxel, since the study did not meet its first endpoint—defined by increase in overall survival—there being some advantage in those cases featuring low expression of ATM telangiectasia mutated) protein measured by IHC [94]. These results confirm that even in a biomarker-enriched population, GC results in a variety and unpredictable pattern of responses in absence of frankly evident drivers. 

### 3.4. miRNAs as Actionable Biomarkers

Strong evidence suggests that alteration in micro-RNA (miRNA) expression acts as important hallmark of cancer [95,96,97]. Expression profiles of miRNAs can distinguish esophageal tumor histology and can discriminate between normal tissue and the transformed one. Moreover microRNA expression might identify patients with BE at high risk for progression to adenocarcinoma [98,99,100]. MiRNA signatures have been investigated in GC for both diagnostic, prognostic purposes as well as to differentiate histologic subtypes and other gastrointestinal cancers [101,102,103,104,105,106,107,108,109]. Thus, miRNA signatures might act as diagnostic and prognostic in upper GI tumors, biomarkers as summarized in Table 3. 

## 4. Tumor Inflammatory and Immune Microenvironment 

The concept of targeted cancer approach has been centered on the neoplastic cells. This paradigm has been now shifted to a more comprehensive understanding of molecular machinery of cancer development which points out the complex interaction between malignant cells and tumor surrounding stroma, which is essential to support each steps of malignant progression [110,111]. This issue is mainly relevant in upper GI cancers in which, on one hand, detection of actionable genetic drivers is rarely reported and on the other, environmental exposure is known to induce inflammatory responses, which ultimately leads to constitutive activation of cellular pro-proliferative and pro-survival signals.

### 4.1. Cancer-Related Immunogenic Cascades

Esophageal cancer cells are considered to display high immunogenicity and can induce massive antitumor immune responses already in the early disease stages. Moreover, all the main cancer-associated risk factors, namely smoking and alcohol, are associated with chronic irritation of the esophageal epithelium and to tissue damage mediated by the consequent production of reactive-oxygen-species (ROS) [112,113]. In addition, changes in the microbiome defined by a relevant decrease of Gram-positive bacteria, are associated to both esophagitis and BE [114] and promote production of lipopolysaccharide (LPS) which in turn, induces inflammation via Toll-like receptor 4 and NF-κB activation. Overall, chronic inflammation activates several cancer-associated signaling pathways [115]. Among them, interleukin 6 (IL-6)/signal transducer and activator of transcription 3 (STAT3) cascades are known to play a relevant pathogenic role in EC. Many different cell types, monocytes, fibroblasts and endothelial cells that reside around the tumor mass produce IL-6. Moreover, EC cells produced both IL-6 and its receptor (IL-6R), thus suggesting that an autocrine/paracrine loop might cooperate in tumor progression and invasion [116,117]. The overexpression of NF-κB (nuclear factor kappa-light-chain-enhancer of activated B cells) transcription factor defines a second mechanism that is known to modulate EC-surrounding microenvironment. Notably, NF-κB is emerging as a potentially effective target since it is involved in regulating cellular apoptosis and angiogenesis [118]. Its main downstream transducers are interleukin-1β (IL-1β) and interleukin-8 (IL-8). The latter, also known as CXCL-8 (C-X-C Motif Chemokine Ligand 8), acts as neutrophil chemotactic molecule and is implicated in the progression of several cancer types, among which EC [119]. Similarly, the activation of IL-1β is associated to tumor growth, chemoresistance and poor patient prognosis [120]. STAT3 and NF-κB converge on several transducers: among them, prostaglandin E, produced by cyclooxygenase-2 (COX-2), which is active in promoting upper GI cancer-related inflammatory reactions and, ultimately, in inducing chemo-resistance [121]. Chronic inflammation is also involved in attenuating anti-tumor immunity, which is orchestrated by several cell populations such as myeloid-derived suppressor cells (MDSCs) and regulatory T cells (Tregs). Expansion of MDSC or immature myeloid cells is modulated by inflammatory mediators (IL-1β, IL-6 and PGE2) [122] and growth factors (i.e., VEGF). These cells can directly inhibit T-cell activation and natural killers (NKs) cytotoxicity, while induce Tregs [123]. The latter are also directly recruited by EC cells through CCL-17 (C-C Motif Chemokine Ligand 1) and CCL-22 (C-C Motif Chemokine Ligand 22) production and by macrophages via the CC-R4 (C-C chemokine receptor 4 receptor) [124]. Moreover, they can derive from the conversion of Th-17 cells when stimulated by TGF-β and IL-6 [125]. Other immune cells, such as tumor-associated macrophages (TAM), display more pro-tumorigenic functions, such as induction of angiogenesis and promotion of malignant cell invasive capacity. TAM expansion with M2 polarization can occur in the presence of Th2-related cytokines. Furthermore, TAMs and malignant cells both express immune checkpoint molecules as PD-L1/2 that can inhibit T cell activation. Indeed, high PD-1/2 expression in EC [126] has been correlated with decreased CD8+ T cell infiltration [127]. The other checkpoint molecule CTLA-4 most often acts as inhibitory receptor on immune cells; however, its expression has been also reported directly in tumor cells [128]. In EAC patients, the upregulation of Th-2 associated cytokine (e.g., IL-4 and IL-13) promotes M2-differentiated macrophage infiltration. In ESCC patients, increased secretion of tumor-derived macrophage chemoattractant protein-1 (MCP-1) results in TAM infiltration [128]. In addition to the above described cells, which overall feature immunosuppressive behavior, another type of stromal element, the cancer associated fibroblasts (CAFs) negatively modulate antitumor immunity in various cancer types, among which EC [129]. CAFs—in EC—can trigger the expression of fibroblast activation protein (FAP) and, in turn, induces the secretion of IL-6 and CCL2 [130] which are involved in creating an immune-suppressive tumor stroma, mainly characterized by M2 polarization of activated macrophages [131]. 

Esophageal cancer and gastric cancers are known to carry many common molecular features, which are, more frequently, shared by EACs and intestinal type of gastric tumors [132]. They derived from the inflammation-metaplasia cascade that occurs in the esophageal epithelium in OAC and in the gastric epithelium in intestinal-type GC. Barrett esophagus and OAC may thus originate from a unique gastric stem fraction, originated from the cardia. Within respect to GC, by matching two key elements, which define the tumor-associated immune milieu, namely the tumor-infiltrating lymphocytes (TILs) and the PD-L1 expression level, four different neoplastic subgroups, emerge with specific and prognostic score. The type I (TILs+ PD-L1+) is defined by adaptive immune resistance, quite opposite the type II (TILs− PD-L1−)is characterized by immune neutrality; type III (TILs− PD-L1+), shows intrinsic induction whereas in type IV (TILs+ PD-L1−) other suppressors might have a role in initiating immune tolerance. [133]. Overall, high expression of PD-L1 associated to CD8+, CD3+ and CD57+ TILs and low densities of FOXP3+ TILs represent favorable prognostic factors [133]. As reported above, an increase in the M2 macrophage component predicts poor prognosis, except for signet ring cell carcinoma and mucinous adenocarcinoma in which it has been associated to a favorable outcome [134]. 

### 4.2. The Role of Extracellular Vesicles

Extracellular vesicles (EV) cooperate in modulating the crosstalk between GC cells and surrounding stroma. They are secreted by several cell types and released to the extracellular space; based on their size they are defined as: exosomes (30–100 nm in diameter), microvesicles (MVs, 100–1000 nm in diameter) and apoptotic bodies (1000–5000 nm in diameter). The smallest type, the exosomes, are nano-sized vesicles, which are enveloped by a lipid bilayer and are, then, secreted from the plasma membrane into the extracellular space. They play an important role in GC onset and progression [14] mainly through overexpression of multiple proteins, miRNAs and LncRNAs [135,136]. Interestingly it has been documented that they actively promote distant growth of neoplastic clones. In detail, Zhang et al. showed that epidermal growth factor receptor (EGFR)-containing exosomes secreted by GC cells can be delivered into the liver, where they were ingested by liver stromal cells. Here, EGFR, by inhibiting miR-26a/b expression, activated hepatocyte growth factor (HGF) [137]. The latter, through a quite paracrine loop, bound its receptor MET expressed on the migrated cancer cells thus triggering the MET-driven invasive growth process [138].

### 4.3. Modulation of Tumor Microenvironment by Ionizing Radiation

In this complex context, the role of ionizing radiation and its interaction with TME emerges as relevant, both locally and under the perspective of its potential abscopal effect. Indeed radiotherapy (RT) represents one of the main treatment strategies in the therapeutic management of oncologic patients, among which those affected by upper GI cancers. Although primarily addressed to kill cancer cells, ionizing radiation also regulates the expression of the different immune cells normally recruited at the periphery of the tumor [139]. Such interactions are likely to impact on tumor growth/dissemination and the capability of a systemic treatment to be particularly efficacious in tumor control. More specifically, radiation, which can be delivered in different doses and treatment fractions, can from one side act as an in situ vaccine leading to immunogenic tumoral cell death; this event is responsible for the release of specific tumor associated-antigen and other molecules (DAMPS) which activates antigen-presenting cells (APC) which ultimately lead to CD8+ Cells activation. Besides, radiation can not only stimulate intratumoral infiltration of macrophages but can also lead to an overexpression of both FGF2 and its receptors (FGFRs). This signaling pathway can switch macrophage phenotype from M1 to M2, which is typically associated to resistance to radiation [140]. Moreover, in EC and GC, increased PD-L1 expression levels have been associated to worse response to ionizing radiation, at least in neoadjuvant setting. The mechanistic explanation of this finding has been related to the overexpression of PD-L1. The latter is directly promoted by the interferon-gamma produced by the CD8+ lymphocytes, through the Janus kinases-Signal Transducer and Activator of Transcription proteins (JAK-STAT) pathway. Notably, high PD-L1 expression has been associated with the induction of the epithelial-to-mesenchymal transition phenomenon is required to tumor distant spreading [141].

## 5. Immunotherapies: Novel Insights and Advances

Systemic treatment of advanced upper GI cancers encompasses combination of multiple lines of chemotherapy, in absence of standard of care regimens. Combinatorial schedules include platinum and fluoropyrimidine doublets, cisplatin/5-fluorouracil (5-FU) or cisplatin/capecitabine. Trastuzumab is associated in HER2-positive cases. Other molecules, such as irinotecan and taxanes, can be associated with fluoropyrimidines and/or platinum or monoclonal antibodies as ramucirumab (a fully humanized molecule directed against vascular endothelial growth factor receptor 2-VEGFR2) or used as monotherapy for unfit patients (for detail see https://www.nccn.org, [142,143,144,145,146]).

### 5.1. Immune Checkpoint Inhibition in Clinical Trials

Immunotherapy with immune checkpoint inhibitors (ICIs) has led to a deep change in therapeutic paradigms of advanced tumors, including that of upper GI cancers. However, until now no validated role for immunotherapy has been approved. About 50% of these tumors express PD-L1 but unlike NSCLC or melanoma, this expression occurs predominantly in the peri-tumor inflammatory stroma while it is minimal on cancer cells [147]. Thus, the specific localization, affects, on one hand, PD-L1 expression as validated biomarker, whereas, on the other, is coherent to the poor responses to ICIs that typically characterize these cancers. Similarly, CTL-4 is considerably expressed in GCs (about 80% of cases but it mostly regards immune stroma cells. Detailed lists of studies evaluating immune checkpoint inhibitors are easily available in literature [148,149]: Table 4 summarizes the first and key clinical trials which evaluate the role of most known ICIs (nivolumab and pembrolizumab) in therapeutic intervention against upper GI tumors. The results from ATTRACTION family trials provide robust evidence for the use of nivolumab in case of first line chemotherapy failure. Overall, they led to the approval of nivolumab as therapeutic option in PD-1-unselected metastatic/recurrent gastric cancer in Asian population (Japan, Taiwan and Korea) [150,151]. The PD-binding monoclonal antibody pembrolizumab has been reported, by the KEYNOTE series trials, to add an advantage in patient outcome when used in advanced disease, mainly in those tumors which overexpress PD-L1. However, in the KEYNOTE061 trial [152], pembrolizumab failed to provide a survival benefit over paclitaxel in advanced GC patients who had progressed after first line treatment with standard chemo. More recently, the novel anti PD-1 antibody, toripalimab, has demonstrated a safe profile and promising antitumor activity in patients with advanced GC alone or in combination with conventional chemotherapy schedules. In this context, the high tumor mutational burden (TMB) emerged as powerful predictive marker of overall survival (OS) [153]. The phase III, randomized JAVELIN Gastric 300 trial has been the first comparing avelumab, an anti-PD-L1 antibody, with chemotherapy in the third-line setting in advanced GC/GEJ cancers. Avelumab failed in improving OS but demonstrated an anti-tumor activity comparable to that of chemotherapy with a more advantageous safety profile [154]. The combination of two different ICIs (anti PD-1 and PD-L1 or anti CTL4) have shown, until now, controversial results. The association of tremelimumab (anti CTLA-4) to durvalumab (anti PD-L1) did not add significant advantages in chemo-refractory GC and GJE cancers [155], whereas the combination of nivolumab and ipilimumab demonstrated more favorable safety and efficacy profiles [156]. Although further investigations are required, combinatorial approaches are now under investigation even in adjuvant settings [157,158]. Among phase III studies, the KEYNOTE-585 trial (NCT03221426) is evaluating perioperative administration of pembrolizumab plus chemotherapy [159] and the Asian ATTRACTION-05 trial (NCT03006705) is comparing S-1/CAPOX (capecitabine plus oxaliplatin) plus nivolumab vs. S-1/CAPOX plus placebo as postoperative approach. Furthermore, two randomized phase II trials are currently ongoing: the DANTE trial (NCT 03421288) evaluating peri-operative use of atezolizumab (anti-PD-L1 antibody) combined with FLOT (docetaxel, oxaliplatin, leucovorin and 5-fluorouracil) [160] and the IMAGINE NCT04062656 randomized, four-arm, chemotherapy-controlled modular trial in subjects with histologically confirmed, resectable GC or GEJ adenocarcinoma. An increase to 35% is estimated to be clinically relevant when patients are treated with either nivolumab in combination with chemotherapy or nivolumab and another immuno-oncology agent (e.g., ipilimumab or relatlimab) [161,162]. Great interest is now addressed to the combination of ICIs and targeted molecules, which seems to be promising although findings are still afar to be conclusive. The combination of durvalumab, targeting PD-L1 olaparib, a PARP (poly ADP ribose polymerase) inhibitor, seemed to be well tolerated in absence of serious adverse event as demonstrated by the phase II MEDIOLA basket trial, which included advanced GCs [163]. Similar results, in terms of safety and clinical activity, have been obtained by adding durvalumab to targeted VEGFR2 inhibitor ramucirumab [164].

Overall, there is an extreme heterogeneity regarding the efficacy of immune checkpoint inhibition in upper GI cancers. It should be noted that published data are highly heterogeneous within respect to disease stage, treatment schedules, different methods of evaluation of PD-L1 expression levels (tumor proportion score (TPS), combined positive score (CPS), different antibodies used for PD-L1 immunostaining, heterogeneity of considered cells (tumor, stroma and immune cells), cut-offs for positivity (1–50%). Moreover, results could be biased by the fact most of the trials have been conducted only in Asia. The findings also reflect the heterogeneity of the patients enrolled in the trials, which led to controversial results concerning the prognostic implications of PD-L1-expressing tumors. From published data however, several issues deserve main special attention. A relevant example regards a meta-analysis of 15 studies (the vast majority enrolling Asian patients) performed by Gu et al. Overall the authors analyzed 3291 patients and a tremendous heterogeneity in PD-L1 IHC positive expression was reported (from 14.3% to 69.4%) mainly as a consequence of the cut-off values used in different studies (ranging between >1% and >50%) [173].

### 5.2. Tumor Mutational Burden as Actionable Targets

As above mentioned, tumor mutational burden (TMB) behaves as effective biomarker for response to anti-PD-L1 treatment in diverse tumor types and in chemo-refractory GCs [157,174]. Accurate TMB measure requires next generation sequencing techniques (NGS), thus surrogate markers are under investigation for routine sample management. Among them, the TGFB family members (TGFB1, TGFB2 and TGFB3) are active transducers in the epithelial-to-mesenchymal transition (EMT) process. Overexpression of TGFB2 has been reported to be positively associated with EMT status and negatively with TMB levels in GC. It affects TMB levels by regulating the DNA damage repair pathways and immune infiltrates, thus suggesting that detection of TGFB2 expression may predict response to ICIs in GC patients [175]. Furthermore, immune checkpoint inhibitors have been used to treat advanced GCs carrying high-frequency microsatellite instability (MSI-H) or mismatch repair defects (dMMR). Microsatellites are short tandem repeats of DNA, which are mostly located in the non-coding genetic or near the chromosome telomeres and their instability defines hyper-mutable phenotype likely caused by defects in mismatch repair (MMR) [176]. The presence of defective MMR genes, which affect about 17–21% of GCs [177] increases the occurrence of somatic mutation to a mean value of more than 1780 compared to 70 changes that can be found in non-defective lesions [178]. It might predict response to anti PD-L1 agents since the occurrence of genetic changes can potentially allow to encode for not-self immunogenic neoantigens. 

### 5.3. Active Immunization Strategies

The above-described results provide a solid rationale for identifying GC patients who may benefit from ICI therapy based on specific tumor genetic asset [179]. In addition, more recent progresses have been reached in the field of tumor immunotherapy. During the past decade, the definition of a strategy to molecularly identify tumor antigens (TA) recognized by immune cells in patients with cancer lead to dramatic progress in tumor immunology. Active immunization is based on the use of an immunogen to generate a host response aimed at eliminating malignant clones in a controlled way. Several strategies have been developed. 

#### 5.3.1. Adoption of Cytokines

A first approach regards the adoption of cytokines (e.g., IFN-γ, IL-10, IL-2) as relevant component of immune response. Indeed, they can directly act on immune cells and modulating their proliferation and signaling against cancer cells. It is well known that cytokines, such as IL-10, are mainly released because of HP-associated chronic inflammation which is implicated in upper GI cancer onset and progression [180]. In this perspective, several ongoing trials are under investigation with both diagnostic and therapeutic purposes. The NCT00197470 study is focused on evaluating the association of the host genetics with the susceptibility to various gastroduodenal disorders, including HP-associated gastric cancer in Japanese population. The study aims at identifying polymorphisms in the IL-1, tumor necrosis factor-alpha (TNF-α) and IL-10 coding genes to clarify the association between those changes and cancer risks to early locate those individuals at higher risks for gastrointestinal malignancies development. Another strategy that is now active in solid cancers among which upper GI tumors regards early detection of cancer recurrence by monitoring changes in a panel of circulating inflammatory cytokines (IL-1, 6, 8, 10, 12 and TNF-α) before and after chemo-radiation (NCT00502502). The phase II randomized clinical trial NCT03554395 compares activated CIK (cytokine induced killer cells) armed with anti-CD3-MUC1 bispecific antibody for advanced GCs to evaluate its safety and clinical efficacy. Another ongoing trial (NCT01783951) has been designed with a similar goal, namely, to evaluate the antitumor effect and safety of activated dendritic cell CIKs (DC-CIK) plus S-1-based chemotherapy for advanced gastric cancer. Interestingly, it has been reported that PD-L1 in human GC inhibits cells to cancer progression and improves cytotoxic sensitivity of cancer cells to CIK therapy [181].

#### 5.3.2. Cancer Vaccines

A second promising strategy is related to cancer vaccination. Cancer is a disease of genes and the occurrence of somatic mutations in oncogenes and tumor suppressor genes drives malignant transformation. However, the accumulation of passenger and driver genetic changes generate cancer-specific neoepitopes that are recognized by autologous T cells as not-self: these molecules on the surface of cancer cells identify ideal targets for vaccines [182]. Great interest in addressed towards clinical development of such therapeutic approach. Well known cancer peptides/proteins recognized by CD8+ and CD4+ lymphocytes are, for instance, melanoma-associated antigen (MAGE-3) [183] and HER-2/neu [184]. Several studies are ongoing. The NCT02276300 study is phase I clinical trial which investigates vaccination against HER2-derived peptide in advanced breast and gastric cancer. BVAC-B is immunotherapeutic vaccine, which uses B cell and monocytes as antigen presenting cell and is under investigation in patients with progressive or recurrent HER2/neu positive GCs (NCT03425773 study). The NCT00023634 trial has been designed to determine toxicity of EGFRvIII peptide vaccine with sargramostim (GM-CSF) or keyhole limpet hemocyanin (KLH) as adjuvant approach in patients carrying EGFRvIII-expressing upper GI cancer. Although not fully documented in upper GI cancers, the variant III of the EGFR receptor seems to behave as oncogene in several solid tumors [185]. Another vaccination strategy aims at using epitope peptide restricted to HLA-A*0201 and a first I trial has confirmed the feasibility and safety of this approach [186]. Subsequent phase II trial is ongoing (NCT00681252). Vaccination using survivin epitope peptide might induce cytotoxic T lymphocytes (CTL) from peripheral blood mononuclear cells of healthy donors. It exhibited specific lysis against HLA-A2 matched tumor cells in vitro and in primary cell cultures derived from GC patients [187]. Vaccination with autologous tumor-derived heat shock proteins (HSPs) is another novel promising approach in GC. The HSP gp96-peptide complexes, as chaperone, can specifically interact with antigen-presenting cells (APCs) and induce their activation. This process allows the secretion of several cytokines and chemokines which, in turn, promote CD4+ and CD8+ T-cell antitumor immune response [188]. This approach resulted safe and advantageous in neoadjuvant settings when combined with conventional chemotherapy in patients affected by les aggressive diseases [189]. Some trials have investigated the use of vaccines against dendritic cells (DCs), which infiltrate tumor stroma. Importantly, the DC density predicts GC prognosis, being higher levels associated to improved OS [190]. An ongoing trial (NCT03185429) aims at learning about the safety and tolerance of autologous TSA-DC cell and evaluates the efficacy and feasibility of the cell therapy compared to standard regimens. Preclinical [191,192] and clinical studies [193,194] have demonstrated that DCs transfected with stabilized mRNA coding for tumor-associated antigen/whole tumor RNA can generate potent anticancer immune responses. In theory, RNA-based vaccines present some potential benefits if compared to classical vaccination approaches: (i) they are pharmaceutically safer, since they cannot integrate with DNA and seem to be active in absence of serious adverse event; (ii) they can target multiple tumor-associated epitopes; (iii) they are not MHC-restricted. However, their clinical application has been limited, until now, by difficulties in obtaining stable and efficient mRNA delivery and a technical improvement is required before fully reaching the clinical scenario [195,196]. More integrated strategies encompass combination of vaccines with standard chemotherapy, which aims at exploiting the above-mentioned potentiality of chemotherapy to upregulate tumor immunogenicity. Notably, a preliminary treatment with conventional chemotherapeutic agents can promote ICI sensitivity, as widely demonstrated in NSCLC [197] and in BRCA1-deficient triple-negative breast cancer models [198]. In adjuvant setting in GC, several combinatorial trials are ongoing. The combination of an adjuvant bacille Calmette-Guérin (BCG) vaccine with chemotherapy can improve OS when compared to chemotherapy alone [199]. Similar results have been obtained with vaccination with gastrin-17 diphtheria toxoid (G17DT)-targeting gastrin peptide combined with chemotherapy [200]. Chemotherapy treatment can sensitize to vaccine against tyrosine kinase receptors, as well. For instance, vaccination using peptides derived from human VEGFR 1 and 2 combined with standard chemotherapy can significantly increase the OS of patients carrying advanced GCs [201]. Preliminary results from vaccination with IMU-13, a structure made of three individual B-cell epitope peptide sequences selected from HER2/neu receptor, plus chemotherapy vs. chemotherapy alone is ongoing on upper GI cancer patients [202]. Finally, attempts of combinations of different novel immunotherapeutic strategies are under investigation. In vitro and in vivo strategies have been adopted to enhance immune response to a low immunogenic tumor cell line obtained from a spontaneous gastric tumor of a CEA424-SV40 large T antigen (CEA424-SV40 TAg) transgenic mouse. In detail, lymphodepletion has been obtained by treating animals with cyclophosphamide and then reconstructed by using syngeneic spleen cells. Subsequently mice underwent effective vaccination with a whole tumor cell vaccine combined with GM-CSF. However, recurrence of Tregs should reduce efficacy of this kind of vaccine in long-term perspective [203]. 

### 5.4. Passive Immunization Strategies

Passive immunization is—by definition—induced artificially when antibodies are given as a therapy to a nonimmune individual. Within respect to cancer, this concept refers to the administration of active humoral immunity in the form of pre-formed antibodies or effector lymphocytes against neoplastic clones. Several approaches are under investigation. 

#### Adoptive Cell Therapy

The most promising approach of passive immunization regards adoptive cell therapy. The latter provides T cells isolated from a patient, manipulated and expanded in vitro and then re-infused into the patient itself [204]. Adoptive cell therapy (ACT) using TILs refers to the passive transfer into a patient of antitumor T lymphocytes which can virtually destroy the tumor mass. Similarly, to active immunization contexts, concomitant treatment with chemotherapy can increase ACT efficacy in GCs. To sustain this hypothesis, it has been shown that oxaliplatin, by stimulating high-mobility group box 1 (HMGB1) protein to induce anti-cancerous T lymphocytes, can promote immune-mediated apoptosis of cancer cells [205]. Several in vitro and in vivo studies on drug-resistant GCs, demonstrated that the combination of alkylating-like agents with CIK cells induces the release of a high amount of cytokines. It seemed that the T lymphocyte reduction obtained by chemotherapy, can improve the efficacy of ACT therapy by stimulating the persistence of endogenous T cells in circulation, in parallel with a reduction of immune reactions in non-transformed organs. However, these encouraging results were associated with the occurrence of severe infectious adverse events and this point seriously limited the clinical development of this strategy. A more promising type of adoptive T cell immunotherapy is related the use of chimeric antigen receptor (CAR) T cells. The latter are synthetic receptors that can re-program T cells. Their signaling domain enables the CAR T cells to activate effector functions and expand upon recognition of antigens on cancer cells [206]. Results from preclinical studies of the clinical use of CAR T cells against upper GI cancers are encouraging, although this approach requires complex technologies. Moreover, an important issue is the identification of the surface antigen. Targeting therapeutic tumor markers, such as HER2, CEA and DF2, have been carried out in basic and clinical studies. The recently designed bispecific T-cell engagers (BiTEs) identify a class of artificial bi-specific antibodies that are made of two single-chain variable fragments (scFv): the first specific for a T-cell (typically CD3) molecule and the second specific for a tumor-related antigen. The novel secretable BiTE, αHER2/CD3, consists of HER2-specific scFv 4D5, CD3-specific scFv OKT3 and flexible linkers can specifically target HER2+ tumor cells, such as those found in gastric cancer and CD3+ human T cells [207]. Folate receptor 1 (FOLR1), also known as folate receptor alpha and folate binding protein, is a glycosylphosphatidylinositol-linked protein is frequently overexpressed on the GC cell surface and it cannot be found in health areas [208]. Both FOLR1-CAR KHYG-1, a natural killer cell line and FOLR1-CAR T cells has been demonstrated to recognized FOLR1-expressing GC cells in a MHC-independent manner: this fact promotes the release of several cytokines and induce cancer cell apoptosis [209]. PSCA, formerly named as prostate stem cell antigen, is a glycosylphosphatidylinositol (GPI)-anchored cell surface protein belonging to the Thy-1/Ly-6 family. Notably, anti-PSCA CAR-T cells exert strong anti-tumor cytotoxicity in vitro and can impair tumor dissemination in in vivo animal models [210]. Interestingly, CAR T cell approach has been exploited also against mesothelin, that is expressed in GC tissue, both in vitro and in vivo with favorable results defined by strong cytotoxicity and significant regression of GC subcutaneous masses [211]. Within respect to esophageal cancer, EphA2 (erythropoietin-producing hepatocellular receptor A2), which is one of the Ephrin receptor family, is a frequently overexpressed surface antigen. CAR-T cells designed against EphA2 induce the secretion of many cytokines and display a dose-dependent capacity of cancer cell death in vitro [212]. Moreover, it is well known that PD-1 can trigger or inhibit signals, which play a main role in the tumor environment, through combining with PD-L1. This combination can not only block the activation of T cells by blocking the first and second T cell signal but can also assist regulatory T cells (Tregs) to play an inhibitory function and induce helper T cells (Ths) convert to Tregs. The widespread presence of immune checkpoints in a variety of solid tumors, among which upper GI cancers, may be one of the main reasons for the poor effect of CAR-T technology in solid tumors. Recent indications show that bi-specific Trop2/PD-L1 CAR-T cells have the high therapeutic potential against GC [213]. Several clinical studies are ongoing. Among them, the combined phase I and II NCT03706326 trial in advanced EC, aims at assessing the safety and efficacy exploiting combination of immune checkpoint blockade and CAR T cells. In detail, the study evaluates and compares the effects of anti- MUC1 CAR T cells alone, anti- MUC1 CAR T combining PD-1 knockout engineered T cells and PD-1 knockout engineered T cells. The efficacy of this approach is now under investigation also in several trials in gastric cancer patients (NCT02862028, NCT03615313 and NCT03182803). 

## 6. Conclusions and Remarks

Although a relevant number of genomic alterations are known to be active in upper GI cancers, few actionable targets can be effectively exploited for diagnostic and therapeutic purposes. Growing evidence suggests that immunotherapy could play a relevant therapeutic role alone and in combination with chemo-radiotherapy and other systemic therapies. Viral infection, mutational burden and MSI status are specific players into constant interconnection between tumor and microenvironment, which modulates response to immune checkpoint inhibitors. The therapeutic landscape is rapidly evolving due to constant refinement and validation of molecular biomarkers. The unique context-related malignant behavior that characterizes upper GI cancers drives responses to novel immune and cell therapies. It remains to be clarified if the genetic and immunological heterogeneity may be somehow related to the different anatomic districts that globally defines the upper GI tract. A deep understating of these processes is challenging and requires a multidisciplinary approach. This will lead—in the near future—to more durable clinical responses in a perspective of full treatment personalization.

## Figures and Tables

**Table 1 cancers-12-02105-t001:** A summary of the morphologic and immunohistochemical profiles of upper GI tumors [18,19,20]. The most common morphologic and immunohistochemical (IHC) traits distinctive of main neoplastic lesions affecting the upper GI tract.

Tumor Type	IHC Markers			Gross Features	
	+	−	+/−	Macroscopic Appearance	Imaging
ESCC	CK5, CK6, CK10, CK14, p40	CK7, CK20	p53, p16 in cases associated to HPV infection	Early cancer - Plaque-like lesions: Small, sessile polyps or depressed lesions Advanced cancer - Luminal constriction (stricture) with nodular or ulcerated mucosa -Polypoid, ulcerative, varicoid, irregular constricting forms	Double-contrast esophagography: best for detection of early cancer CT: Useful for staging. Mediastinal and abdominal lymphadenopathy and metastases PET/CT: superior to CT in detecting regional and distant metastases Endoscopic ultrasonography (EUS): best technique for determining locoregional extent of tumor
EAC	CK7, CK8, AMACR, weak focal CDX-2	p40, p16, ER, GATA 3, TTF-1	CK20		
Esophageal small cell carcinoma	Chromogranin A, NSE, Synaptophysin, CD56, CK8	p40	TTF-1		
Gastric adenocarcinoma	CK8, CK7	TTF1, p40, ER, p16, MUC1, E-cadherin (Poorly cohesive)	CK20, CDX-2, MUC1, MUC2, MUC5AC	Polypoid or circumferential mass with no peristalsis through lesion (at fluoroscopy)	Best imaging tool: Double-contrast barium study, CT, EUS

SCC = squamous cell cancer ER = estrogen receptor TTF-1 = thyroid transcriptional factor-1 NSE = neuron specific enolase.

**Table 2 cancers-12-02105-t002:** Clinical, anatomic, pathologic and imaging characteristics of upper GI cancers. Main clinically relevant features of esophageal and gastric cancers derived from already available literature data [31,32,33,34,35,36,37,38,39,40,41,42,43,44,45,46].

Features	Esophageal Cancer		Gastric Cancer
	Squamous Cell Cancer	Adenocarcinoma	Adenocarcinoma
Geographic distribution	Eastern Asia	United States and certain European countries	East Asia, Eastern Europe, Central, South America
Smoking history	✓	✓	✓
Other associated conditions		- Obesity - Gastroesophageal reflux disease (GERD)	- Obesity - Socioeconomic position (minors, fishermen, machine operators, nurses, cooks, launderers, dry cleaners) - Gastrectomy
Dietary factors	Low apport of fruits and vegetables leading to low antioxidant levels and vitamin deficiencies	- Red meat and processed food items - Protective role of raw fruits and vegetables and dark-green, leafy and cruciferous vegetables, carbohydrates, fiber, iron	- Alcohol - Protective role of fresh fruits and dark green, light green and yellow vegetables rich in B carotene, vitamin C, E and foliate
Histology and Anatomic localization	Squamous lining of middle esophagus	- Glandular differentiation featuring tubular, tubulo-papillary or papillary pattern of growth. - Distal part of esophagus	- Diffuse type: poorly cohesive, including signet ring histotype - Intestinal type: - papillary - tubular - Proximal stomach near the junction - Distal stomach (intestinal type)
Endoscopic features	Polypoid masses, flat or ulcerated	- Mucosal irregularities, which might be associated to ulcerated or infiltrative lesions - Exophytic masses which can obstruct the lumen	Early GC (EGC): - elevated - superficial ➢ superficial elevated, ➢ superficial flat ➢ superficial depressed - depressed. The most common lesions of EGC were usually manifested by erythema and erosion.
Oncogenic viruses	*Human Papilloma Virus* (HPV): role not well established	*Helicobacter pylorii* (HP) infection is inversely correlated	- *Helicobacter pylorii* (HP) - Epstein-Barr virus (EBV) associated to GSC
Chronic inflammation	Achalasia		
Premalignant lesions	Squamous dysplasia	Barrett’s esophagus (BE)	- HP-related chronic atrophic gastritis - Intestinal metaplasia and dysplasia - Early GC (10% of diagnosis)
Variants and differentiations	- Basaloid squamous cell carcinoma - Verrucous carcinomas - Spindle cell carcinoma - (or carcinosarcoma)	- Mucinous - Signet ring cell	- Mucinous - Mixed

**Table 3 cancers-12-02105-t003:** MiRNA expression in esophageal and gastric cancers. In each case, expression has been associated with its functional (diagnostic and/or prognostic) value based on literature reports (PubMed search according to the following keywords: esophageal/gastric cancer & miRNA).

Cancer Type	Diagnosis	Prognosis
Esophageal Cancer	Expression in transformed tissues, not in normal areas: miR-21, -34a, -205, -203, -93, -375, -494, -29c, -148, -203 Role in tumor onset: miR-4286, -502, -374a Expression in pre-neoplastic lesions: miR-144, -155 Increased cancer risk: miR-196a2, -146a, -423 Association with tumor regression: miR-192, -194	Tumor cell proliferation and invasive phenotype: miR-26a-5p, -195, -338, -200a-3p, -196a, -486-5p, -218, -503, -374a, -183, -150-5p Inhibition of cell proliferation and migration: miR-652, -124, -485-5p, -139-5p, -203-3p, -21-5p, -155 ↑ Tumor progression and metastatic capacity: miR-4319, -451, -1207-5p, -143-5p, -3687, -6743-5p, 20b ↓ Patient survival: miR-1301-3p, -431-5p, -769-5p, -451 ↓ Metastatic potential: miR-124, -210, -491, -140 Resistance to chemotherapy: miR-193, -141-3p, -27, -96 Sensitivity to chemotherapy: miR-218 Radioresistence: miR-24, -133a, -96 Radiosensitivity: miR-124
ESCC	Expression in transformed tissues, not in normal areas: miR-17-, -19a, -7, -1297, -196a, -613, -143, -122, -302b Role in tumor onset: miR-373, -153-3p, -145-3p, -449a-5p, -483-5p, -455-3p, -100, -181a: Increased cancer risk: miR-423, -196a2, -499, -219-1	↑ Cancer progression and metastatic capacity: miR-124, -130a-5p, -196a, -214, -23b-3p, -370, -129, -31, -548k, -612, -30b, -146a-5p, -92b, -483-3, -425, -1290, -192, -503, -195, -183 ↓ Tumor progression: miR-33a-5p, -384, -133a-3p, -615-3p, -120-3p, -196a, -126, -30d, -199a-5p, -338-3p, -203 Tumor cell proliferation and invasive phenotype: miRNA-141, -10b-3p, -365, -424, -1470, -214, -503, -375, -18, -101, -889, -208, -16, -518b ↓ Tumor cell proliferation: miR-133b, -338-5p, -10a, -6775-3p, -125b, -302a, -1, -99a, -26, -100, -34a, -150, -383, -186, -1291, -106a, -129-2, ↑ survival: miR-30e, miR-124, -874-3p, -502,-335 ↑ Cell migration and invasion: miR-548-3p, -576-5p,-25,-99b,-375,-106b, -630, -675, -373, -200b,-25, -205,-92a ↓ Patient survival: miR-145, -191, -138, -1469, -574-3p, -625, -382, -17, -18a, -19a, -150, -486-5p: ↓ Cell survival, migration and invasion: miR-145, -202, -92b, -328, -204, -520g, let-7g, let-7i, -218, -101, -217, -494, -508, -429 ↑ Radiosensitivity: miR-27a, -136, -339-5p, -193b,-338-5p,-381, miR-22 Sensitivity to chemotherapy: miR-145, -125a-5p, -214-3, -449, -218 Resistance to chemotherapy: miR-24, -455-3p, -483, -214, -141
**EAC**	Higher expression in EAC (vs ESCC): miR-148a, -29c Early tumor onset: miR-92a-3p, -223, -31, -375, -192, -196a, -203a, -130,-663b, -421, -502-5p, -1915-3p, -601, -4286, -630, -575, -494, -320e,-203, -625-3p, -21, -31, -192, -194, -200a, -194 Low expression in BE evolving to EAC: miR-153, -192, -194, -194-3p, -200a, -215, -133b, -203, -205,-143, -145, -31, -31-3p, -375, -143, -145, -215	Tumor progression and invasive phenotype: miR-196, -145, -17, -19a/b, -20a, -106a, -330-5p, -99b, -199a-3p, -199a-5p Tumor recurrence: miR-331-3p Better prognoses and ↑ patient survival: miR-100-5p, -133b, -302c, -222 ↓ Patient survival: miR-126, -125a, -15, --375(lower): Chemoresistance: -221, -187 Chemosensistivity: mir-330-5p, -148 Increased expression after chemotherapy: miR-222, -549 Radiosensitivity: miR-31: Pathologic response (low expression): miR-505 *, -99b, -451, -145 ↓ Patient survival: miR-375, -31, -21
**Gastric Cancer**	Higher expression in cancer (vs normal tissue and gastritis): miR-19a-3p, -22-3p, -146a-5p, -483-5p, -421, -29b-1-5p, -27b-5, -10b, -21, -93, -107, 124, -20a, -22, -10b, -21, -93, -107, -106 a High expression in health gastric mucosa (vs cancer): miR-26a, -375, -1260, -26a, -142-3p, -148a, -195, -545 High expression in gastritis (vs cancer): miR-146a, -155 Histologic differential expression: miR-200c, intestinal-type: miR-32, -182, -143, 520c, -229-5p; signet ring: miR-99a-5p, Lauren differentiation: 193b Expression in pre-invasive areas: let-7i-5p, miR-146b-5p, -185-5p, 22-3p Early cancer onset: miR-101-3p, 106a, -9 17-92, 223, -324-3p HP-associated cancer: miR-17-3p, -17-5p/3p, -222-3p, -143-3p Cancer susceptibility: miR-993 Ectopic expression: miR-143, -145, -4290: significant impact on tumor growth	↑ Cancer progression and metastatic capacity: miR-21,-125b, -199a, -100,-34a, -146a, -335, -301a, -224-5p, -92a, -136, -106, -129, -215, 423-5p, iR-181a-5, -28, -26a, -155, -589, -142-3p, -23a, -658, -491-5p, -4284, -200, -634, -196b-5p, -135b-5p, -638, -155, -93-5p, -204, -211, -93-5p, -144,-229-5p,- 425-5p, -221,-222,-497,- 146a, -15b-5p,-182-5p,-425-5p, -1258, -551b, -491-5p,-532,-132-3p,-423-3p, -3622b-5p, -187, -1296-5p, -574-3p, -520b/e, -376c-3p, -330-3p, -187,-501-5p, -107,-125b,-221-3p, -558,-135a, -483-5p,- 224, -214, -222, -218, -224,-363, -935, -371-5p,-183,-500, -181a, -221-3p, -93-5p, -1296-5p, -663, -508-5p, -96-5p, 32-5p,-373, -153, miR-29c, -124, -135a,-148a, -892a, -20b, -451a, -130a, -398,- 192,-215, -23b-3p, -130a-5p -181a, -18a/19a, -429,-34a, -588,- 543,-885-5p, -153, - 452, -216b, -92a, -675, -223-3p, -214, -93-5p, -23a, -761,- 424-5p, -520c, -101, -425-5p, -203, -638, -15b-3p Tumor recurrence: miR-590-3p Suppression of malignant development: miRNA-339-5p, -129-5p, -139-5p, -489, -520f-3p, -143, -148a-5p, -539-3p, -129-3p, -197, -410,-345,-100, -2195p, -133b, -378, -204, -338, -141, -663, -449a, -376c-3p,-135a, -223, -371-5p, -214,- 630, -539, -218, -202-3p, -16, -1292-5p,- 5590-3p,-155-5p, -361-5p, -449c, -129-5p, -518, -483, -198, -1236-3p, -338-3p, -337-3p,-107,-551b, -138, -204, -29a-3p, -495, -223-5p, -148b-3p, -338, -125a-5p, -585, -148a, -491-5p, -519d-3p, -122-5p, -188-5p, -708, -122-5p, -429, -100, -630,- 203a, -143, -199a/b-3p,- 454, -204, -152, -200a, -302b, -373, -185, -3174, 582-5p, -377, -216a,-361-5p, 142-5p, 329, -197, -599, -130a-3p, 937, -454, -129, -802, -143,-145, 381, -154, -4317, -519d, -31, -124, -584-3p, -140-5p, -154, -302b-3p, -26a/b, -143, -206, -455, -379, -320a, -613, -30c-5p, -944,-30a-5p, -211, -138, -31, -218, -646, -508-5p, -133b, -455,-429, -2392, -195,- 217, -129-5p, -1228, -181, - 509-3-5P, -584-5p, -135a, 134, -101-3p, -381, -29c, 495, -15a-3p, -16-1-3p, 144-3p, 133a, -1296-5p, -647, -224, -644a, -219-5p, -494, -194, -337-3p, -494, -326, -561, -509-3p, -133b, -218, -208a-3p, -1248, -19b, -520f, -203, -18, -370, -200b, -205, -193b, -524-5p, -203, -448, -144-3p, -133a, -1296-5p, -31, -145, -2392, -143, -206, 302b-3p, -1296-5p, -429, - 577, -129-5p/3p, -330-3p, -524-5p, -3174, -139-5p, -375, -32, -485-5p, 1915-3p, -16, -198, -12129, - 876-5p, 105, -1284, -miR-155-5p, 25-3p, -503, -629, 449c, -125a-5p, 127, -331, 1224, -142-3p, 491-5p, 339-5p, 129-3p, -519d, 944, -206,- 411, -4316, -539-3p, -671-5p, -139-5p183, -503, -551b, -99b-3p, -449a, -505, -129-5p, -93-5p, -429, -132, -874, -493, -124-3p, -135a, -206, -148a, -621, -337-3p, -211, -429, -203a, -761, -19b-3p, -6852, -598, - 884-5p, -520a-3p, 140-5p, -1236-3p, -489, -100-3p, -140-5p, -4268, -618, -1297, -378, -216b, 38 Increased cell proliferation and motility: miR-425-5p, -330-3p, -99a-5p, -216b, -638, -17, -4513, -374a, -761, -181a, -647, -217,-144, -23a/27a/24-2, -425-5p,-592,- 374b,-208a, -103a-3p, -423-5p, -340, -136, -615-3p, -28, -93, -214, -205, -23a/b, -16a-3p, -130a, -105, -744, -215, -370, - 215, -103, -196a-5p, -224, -186, -17-5p, -490-3p, -23a/27a/24-2-, 96-5p, -638, -1269, -200c, -3619, -421, -320a, -192-5p, -181a, -148a-3p, -145, let-7 Chemoresistance: -103, -107, -508-5p, -23b-3p, -590-5p, -13147,- 5-5p, 195-5p, -17, -20a, 21-5p, -125b, -200, -145, -132, -939, 129, 99a, -491-135a, -424-5p, -1284, -135b, -17-5p, -765,- 522, -106a Chemosensitivity: miR-223, -200c, ↓-21, -16, - 494, ↓135b-5p, -21-5p, -939, -623, -429, -204, -124 or -3-494, -1,-200, -542-3p, - 320a7, -101, 34a, -33b-5p, -495, -524-5p, -30a, -149, -590-5p, -375, -92a, 375, -362-5p, -7, -192-5p, -613, -590-5p, -218 Pro-angiogenic effect: miR-574-5p, -616-3p Anti-angiogenic effect: miR-218, Radiosensitivity: miR-196b, -190 ↓ Patient survival: miR -486-5p, -552, -647, -519a, -126, -532-5p, -125b, -204, -539, -22, -141, -31, -185, -1297, -19a-3p, -1298, -375, -338-3p, -203, -490-3p, -144, -302a, -302b, -302c, -204, -485-5p, -29c, -124, -135a, -148a, -198, 92a, -1258, -519a, -141, -3923, -29c-3p, 193b, -155 (inverse correlation with tumor stage)

**Table 4 cancers-12-02105-t004:** Main clinical trials evaluating ICI in upper GI cancers [165,166,167,168,169,170,171,172].

Study	Design and Phase	ICI	Cancer Type	Population	Endpoint	Results
ATTRACTION-1	open-label, single-arm, II	nivolumab	ESCC refractory or intolerant to standard chemotherapies.	65 Japanese pts	Safety, efficacy	Positive
ATTRACTION-2	randomized, double-blind, placebo-controlled, III	nivolumab	unresectable advanced or recurrent G/GEJ cancer refractory to, or intolerant of standard chemotherapy	49 pts (Japan, South Korea, Taiwan)	OS	Nivolumab group median OS: 5.26 months vs. 4.14 months in the placebo group. The 12-month OS was 26.2% with nivolumab and 10.9% with placebo
ATTRACTION-3	multicenter, randomized, open-label, III	nivolumab vs. chemotherapy	unresectable advanced- recurrent ESCC (regardless of PD-L1 expression)	419 pts (210 nivolumab vs. 209 chemotherapy)	OS	increased OS (median OS 10.9 vs. 8.4 months)
ATTRACTION-4	multicenter, randomized, open-label, II	nivolumab + S1 +SOX or capecitabine	unresectable advanced or recurrent HER2-negative G/GEJ cancer	40 randomized pts, 39 (nivolumab plus SOX, 21; nivolumab plus CapeOX, 18) and 38 (21 and 17)	Safety, efficacy	Well tolerated. ORR 57.1% with nivolumab plus SOX and 76.5% with nivolumab plus CapeOX. Median PFS 9.7 months and 10.6 months.
KEYNOTE-012	multicentre, open label, 1b	pembrolizumab	PD-L1–positive advanced G/GEJ adenocarcinoma	39 patients	Safety, objective response	13% pts grade 3/4 treatment-related adverse events. 22% ORR
KEYNOTE-059	global, open-label, single-arm, multicohort, II	pembrolizumab	previously treated G/GEJ cancers	259 pts	Safety, efficacy	ORR: 11.6% complete response: 2.3% serious adverse events: 0.8%
KEYNOTE-180	open-label, interventional, single-arm, II	pembrolizumab	metastatic ESCC, EAC that progressed after 2 or more lines of therapy	121 pts	ORR	ORR: 9.9% among all patients, median duration of response not reached
KEYNOTE-181	open-label, randomized, III	pembrolizumab vs. investigator’s choice chemo as second-line therapy	advanced/metastatic ESCC and EAC or Siewert type I GEJ adenocarcinoma	628 and 123 pts in the global and China cohorts.	OS in the ITT, and PD-L1 CPS ≥10 populations.	Pembro and chemo showed comparable OS. Pembro: showed favorable OS in ESCC and CPS ≥10 groups in the global cohort and in all groups in the China cohort. Pembro showed favorable safety in both cohorts
KEYNOTE-061	randomised, open-label, III	Pembrolizumab vs. paclitaxel	Advanced GCs, progressed after first-line treatment with fluoropyrimidine and platinum	592 pts (30 Countries)	OS, PFS in PD-L1 CPS > 1	Failure. Median OS: 9.1 months with pembro vs. 8.3 months with paclitaxel. Median PFS 1.5 months with pembro and 4.1 months with paclitaxel
KEYNOTE-062	III	Pembrolizumab or pembrolizumab + chemotherapy	advanced gastric or GEJ adenocarcinoma			non-inferior/better to chemotherapy in PD-L1
	II	Tremelimumab	second-line treatment in advanced EAC -GC	18	Safety, clinical efficacy, immunologic activity	Most drug-related toxicity was mild; 1 death due to bowel perforation. Four patients SD with clinical benefit; 1 pt PR after 8 cycles

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
