# Peer review of "From Interconnection between Genes and Microenvironment to Novel Immunotherapeutic Approaches in Upper Gastro-Intestinal Cancers—A Multidisciplinary Perspective"

_cancers, 2020, doi:10.3390/cancers12082105_

Round 1

Reviewer 1 Report

The article by Accordino describes "From interconnection between genes and microenvironment to novel immunotherapeutic approaches in upper gastrointestinal cancers. A multidisciplinary perspective" reviewed epidemiology, symptom, diagnosis, and therapy of esophageal and gastric malignancies.  They reviewed the recent development of checkpoint inhibitor therapy and active and passive immunotherapies. The references include 2020 publications and comprehensively covered subjects. 

I only have minor comments that help to navigate the readers

1) I recommend adding the subheadings to the following sections;

line 209 Gastric cancer

line 291 Tumor inflammatory and immune microenvironment

line 389 Immune checkpoint inhibition

line 472 Active immunization strategies

line 552 Passive immunization strategies

2) There are a few typos. Please fix them. 

3) There two of Table 3 in Supplemental tables. 

Author Response

Comments and Suggestions for Authors

The article by Accordino describes "From interconnection between genes and microenvironment to novel immunotherapeutic approaches in upper gastrointestinal cancers. A multidisciplinary perspective" reviewed epidemiology, symptom, diagnosis, and therapy of esophageal and gastric malignancies.  They reviewed the recent development of checkpoint inhibitor therapy and active and passive immunotherapies. The references include 2020 publications and comprehensively covered subjects. 

We really thank the Reviewer for careful revision of our work, which is now better in terms of quality and scientific message. Below the point-by-point answers (A).

I only have minor comments that help to navigate the readers

  • I recommend adding the subheadings to the following sections;
  •  
  1. We agree and really thank the Reviewer for this suggestion and the following sub-headings have been added

line 209 Gastric cancer

  • Genetic features
  • Targeted-based therapeutic strategies
  • miRNAs as actionable biomarkers

line 291 Tumor inflammatory and immune microenvironment

  • Cancer-related immunogenic cascades
  • The role of extracellular vesicles
  • Modulation of tumor microenvironment by ionizing radiation

line 389 Immune checkpoint inhibition

  • Immune checkpoint inhibition in clinical trials
  • Tumor mutational burden as actionable targets

line 472 Active immunization strategies

  • Adoption of cytokines
  • Cancer vaccines

line 552 Passive immunization strategies

  • Adoptive cell therapy

  • There are a few typos. Please fix them. 
  1. A) We really thank the Reviewer for the careful revision of the manuscript. The text has been corrected.

3) There two of Table 3 in Supplemental tables. 

  1. A) We really thank the Reviewer for the careful revision of the supplementary data and table numbering has been corrected.

Reviewer 2 Report

In This review article:  « From interconnection between genes and microenvironment to novel immunotherapeutic approaches in upper gastro-intestinal cancers. A multidisciplinary perspective » the authors described diagnostic and therapeutic updated aspects of esophageal and gastric cancers.

It is a very well structured Review however some Minor points must be addressed.

1- since the authors focus on gastric and esophageal cancer ( adenocarcinoma and SCc) it is not pertinent to discuss differential diagnosis of gastric and esophageal Tumors ( GIST, leiomyoma, lymphoma, and other benign and malignant tumors...) there are too many distracting information about these entities and their diagnosis in the second paragraph (lines 108-129)

2-The table 1 could focus on gastric and esophageal adenocarcinoma and SCC

3- In table 2 the histologic subtypes of gastric  adenocarcinoma are lacking.

4- In line 67 :In the sentence «  However, most of GC diagnoses are performed in III or IV disease.... » it lacks the word « stage » before « III or IV »

5- Why references are indicated in roman Number characters Instead of Arabic numbers ?

Author Response

In This review article:  « From interconnection between genes and microenvironment to novel immunotherapeutic approaches in upper gastro-intestinal cancers. A multidisciplinary perspective » the authors described diagnostic and therapeutic updated aspects of esophageal and gastric cancers.

It is a very well structured Review however some Minor points must be addressed.

We really thank the Reviewer for careful reading of the manuscript and the constructive remarks. We have revised the manuscript accordingly and updated the reference section. Below the point-by-point answers (A).

1- since the authors focus on gastric and esophageal cancer ( adenocarcinoma and SCc) it is not pertinent to discuss differential diagnosis of gastric and esophageal Tumors ( GIST, leiomyoma, lymphoma, and other benign and malignant tumors...) there are too many distracting information about these entities and their diagnosis in the second paragraph (lines 108-129)

2-The table 1 could focus on gastric and esophageal adenocarcinoma and SCC

  1. We thank the Reviewer for these fruitful comments, and we have modified the manuscript and the table 1 accordingly.

3- In table 2 the histologic subtypes of gastric adenocarcinoma are lacking.

  1. We thank the Reviewer for pointing out this criticism and the Table 2 has been integrated by histologic subtypes of gastric adenocarcinoma.

4- In line 67 : In the sentence «  However, most of GC diagnoses are performed in III or IV disease.... » it lacks the word « stage » before « III or IV »

  1. We thank the Reviewer for this careful revision and the word “stage” has been added.

5- Why references are indicated in roman Number characters Instead of Arabic numbers ?

  1. We thank the Reviewer for this critical issue and reference format has been revised and corrected.

This manuscript is a resubmission of an earlier submission. The following is a list of the peer review reports and author responses from that submission.